# Metagenomic, metabolomic, and sensorial characteristics of fermented *Coffea arabica* L. var. Castillo beans inoculated with microbial starter cultures

Maria A. Madrid-Restrepo,[1,2] Ana M. León-Inga,[3] Aida Esther Peñuela-Martínez,[4] Mónica P. Cala,[3] Alejandro Reyes[1,2]

**ABSTRACT** Coffee is one of the most important and widely consumed drinks around the world, and fermentation plays a pivotal role in shaping its quality. This research explores the impact of co-fermentation with "starter cultures" on the sensory and metabolic profiles, as well as on the dynamics of microbial communities involved in coffee processing. Freshly harvested *Arabica* coffee beans were subjected to two wet-fermentation processes, one inoculated with a microbial starter culture and the other undergoing spontaneous fermentation. Quantitative descriptive analysis revealed that the inoculated coffee outperformed the spontaneous fermentation in all sensory attributes, boasting higher sweetness, reduced acidity and bitterness, and the presence of consumer-preferred notes. Untargeted metabolomic analysis identified over a hundred differential metabolites distinguishing both fermentation processes in green and roasted beans. Inoculated coffee displayed elevated levels of compounds such as sucrose, mannitol, methyl phenylacetate, and organic acids like malic, citric, and quinic acid, compounds likely associated with improved sensory perception. The inoculated process was characterized by shifts in the abundance of lactic acid bacteria and *Kazachstania* yeasts, groups linked to desirable metabolites such as lactic, acetic, isobutyric, and hexanoic acids. Our results strongly suggest that the use of starter cultures can enhance coffee beverage quality, as reflected by standardized cupping, metabolic profiles, and microbial community dynamics. Future studies should focus on disentangling microbial contributions and metabolite pathways to inform the design of commercially viable starter cultures for coffee fermentation.

**IMPORTANCE** Our study demonstrates that inoculating coffee fermentation alters the sensory qualities of coffee and reshapes the dynamics of bacterial and fungal communities during this process. We identified distinct changes in microbial diversity and metabolite composition associated with inoculation, which correlated with improved sensory attributes. In addition, we detected aminophenol and phenol at higher levels in spontaneously fermented coffees, compounds that are likely responsible for phenolic defects. To our knowledge, this is the first report directly linking these compounds to defective flavor notes in coffee. Together, these findings show that inoculation not only enhances desirable flavor profiles but may also serve as a strategy to reduce the risk of cup defects by modulating the fermentation microbiota. Our work advances the understanding of community-level microbial processes in coffee fermentation and opens opportunities for developing techniques to produce coffee with unique, high-quality, and reproducible sensory characteristics.

**KEYWORDS** coffee, starter culture, co-fermentation, sensory profiling, lactic acid bacteria, *Kazachstania*, untargeted metabolomics

Address correspondence to Alejandro Reyes, a.reyes@uniandes.edu.co.

The company Café Fino S.A.S. partially financed the research and supplied the samples. However, the company played no role in the experimental design, data generation, or analysis and interpretation of results.

offee is one of the most consumed beverages globally, with over 2 billion cups consumed daily (1). It is a brewed drink prepared from roasted coffee beans, the seeds of plants from the genus *Coffea*. After harvest, seeds undergo fermentation, drying, and dehulling and are then exported worldwide. Following significant annual increases in coffee production (2), coffee prices have reached an all-time high of over 2 dollars per pound (3). This price is closely related to the quality of the coffee bean, which depends on various factors, some intrinsic to the plant and growing conditions and others related to post-harvesting processes. Consequently, some coffee growers have focused on producing higher-quality coffee with specialty sensory characteristics through innovative processes.

One critical post-harvest process that develops the sensory characteristics of coffee is fermentation. At production sites, two processes are typically used: dry and wet processing (4). Dry-processed coffees are fermented as whole fruits, while wet-processed (washed) coffees are fermented without the fruit that surrounds the bean. Regardless of the fermentation type, the objective is to degrade mucilage—a sugary layer protecting the bean—composed mainly of water (85%), sugars, and polysaccharides (5, 6). This layer is degraded by pectinolytic enzymes produced by microorganisms. Fermentation is a complex system of reactions caused by various species of yeast and bacteria that transform fermentable sugars into $CO_2$, ethanol, and aromatic compounds affecting the coffee's sensory profile. Over 150 bacterial and fungal species have been identified as active participants in this process (7, 8), including a cohort of lactic acid bacteria (LAB) (*Leuconostoc*, *Lactococcus*, and *Lactobacillus*), acetic acid bacteria (*Acetobacter* and *Gluconobacter*), g (*Enterobacter*, *Klebsiella*, and *Erwinia*), and certain genera of yeasts (*Torulaspora*, *Pichia*, and *Starmerella*) (9–12).

Alongside microorganisms, a dynamic environment of metabolites significantly alters the flavor, aroma, and overall quality of the coffee. These metabolites include desirable compounds such as acids, alcohols, pyridines, aldehydes, and furans, all of which are primarily produced by yeasts involved in the fermentation process (7, 13). Metabolites commonly found during coffee fermentations include simple carbohydrates (glucose, sucrose, and fructose), certain organic acids (lactic, acetic, citric, and malic), sugar alcohols (arabitol, glycerol, mannitol, and sorbitol), and some low-molecular-mass volatiles (acetaldehyde, ethanol, ethyl acetate, ethyl lactate, and isopentyl acetate) (9, 14, 15).

Furthermore, certain coffee producers have discovered that a carefully managed and supplemented fermentation can positively impact beverage quality. Drawing from the wine, beer, and dairy industries, many coffee producers use inoculants during fermentation to ensure higher quality. Their positive effect on the coffee cup quality has been described. For example, coffee beans fermented with a consortia of *Saccharomyces cerevisiae*, *Lactobacillus plantarum*, and *Bacillus sphaericus* improved physical, chemical, and sensory profiles compared to coffee fermented following a standard process (16). Another study reported that inoculating Arabica coffee with two different species of yeast (*S. cerevisiae* and *Torulaspora delbrueckii*) modified the sensory profile of the resulting beverage and increased its quality score by five points compared to a non-inoculated coffee (15). Compared to non-inoculated coffees, inoculated coffees have higher overall scores for flavor and aroma that cannot be achieved otherwise (8, 17).

The concept of controlled fermentations, involving specific yeast species and strains for inoculation, has been previously explored (17–19). Recently, there has been growing interest in non-standardized inoculations, as producers experiment with various approaches to enhance flavor profiles and improve quality. Over the past decade, producers have begun introducing a wide range of non-traditional inoculants into coffee fermentations. These include fruits, spices, and other materials that serve as sources of diverse microbial communities, as well as established microbial strains originally used in bread or beer fermentations (20–24). These practices aim to leverage complex interactions between microorganisms to create unique flavors increasingly sought after by consumers and specialty coffee markets. For instance, some producers in countries

like Colombia and Costa Rica have successfully utilized so-called "co-fermentation" techniques by incorporating starter cultures to the fermentation process, which not only enhances fermentation efficiency but also enriches the sensory qualities of the final product (25). And although the popularity of non-standardized "co-fermentations" is increasing, it is still not well understood how these unstandardized inoculums are impacting the microbial and metabolite profiles of coffee fermentations, and what the final effect on the coffee product is after fermentation and post-harvesting processes.

This study investigates the microbial communities and metabolic profiles of *Coffea arabica* L. var. Castillo beans were subjected to two distinct post-harvest fermentation processes: spontaneous fermentation and co-fermentation with a mixed starter culture. Our aim is to understand how inoculation perturbs the existing fermentation microbiota and how these community-level shifts translate into differences in metabolite composition and sensory quality. Starter cultures provide both exogenous microorganisms and additional nutrients, which together may alter microbial dynamics, metabolite transfer between the fermentation liquid and the beans, and ultimately the sensory attributes of the final beverage. Because the effects of such interventions are still not fully understood in coffee processing, our work contributes to clarifying how co-fermentation strategies influence microbial succession, chemical transformations in green and roasted beans, and the resulting cup quality.

## MATERIALS AND METHODS

### Fermentation sampling

Coffee bean fermentation samples were collected from a farm located in Risaralda, Colombia (4°55′31.0″N, 75°40′27.8″W). The coffee farm is at 1,600 m of altitude and produces "washed coffee" in accordance with a traditional wet process method, as well as a fermentation method where starter cultures are added at the beginning of the process, developed by the producers.

For the spontaneous wet coffee fermentation (hereby referred to as Spontaneous Wet [SW]), 155 kg of freshly harvested coffee fruits (*C. arabica* L. var. Castillo) were placed in 200 L plastic tanks. Fifty-five liters of water were added, and a spontaneous fermentation was allowed to occur for 100 h. For the wet process with the starter culture (hereby referred to as Inoculated Wet [IW]), the same amount in weight of coffee fruits was submerged in 30 L of fresh water. Additionally, 10 L of a starter culture and 30 kg of crushed fresh fruit (pineapple, *Ananas comosus*) were added to the tanks, following the farm's typical production process of this type of fermentation. The starter culture consisted of 1.36 kg of wheat flour, 2 kg of raw sugar, 60 L of water, and 20 g of a defined cocktail of two commercially available *S. cerevisiae* strains ($1.0 \times 10^{10}$ cfu/g). Strains of added yeast are not detailed due to a verbal confidentiality clause agreed previously with the coffee producers. The coffee-starter culture mixture was left to ferment for 100 h, and the pH was monitored to ensure it stayed between 5.5 and 6.0. For metagenomic analyses, samples of 50 mL of the liquid fraction of the fermenting coffee at 0, 50, and 100 h were collected in triplicate from independent tanks so that an early sampling does not disturb the fermentation process of later samples. During 50 and 100 h, samples of the liquid fraction were collected from the top and the middle of the barrel without disturbing the contents to reduce bias toward aerobic or anaerobic microorganisms. The collected samples were flash frozen in a dry ice-ethanol bath (−78°C). After the fermentation process, the fermented coffee beans were washed and sun-dried for 15 days to obtain dried coffee parchment (dcp) with a moisture content between 10% and 12% wet basis. At this time point, 100 g of the dcp beans were collected for metabolomic analyses (green coffee). Afterward, the remaining dcp beans were dehulled and roasted following the procedure described below. An additional 100 g of the roasted coffee beans was also collected for metabolomic analyses (roasted coffee). All samples were flash frozen after collection in a dry ice-ethanol bath and transported to the lab, where

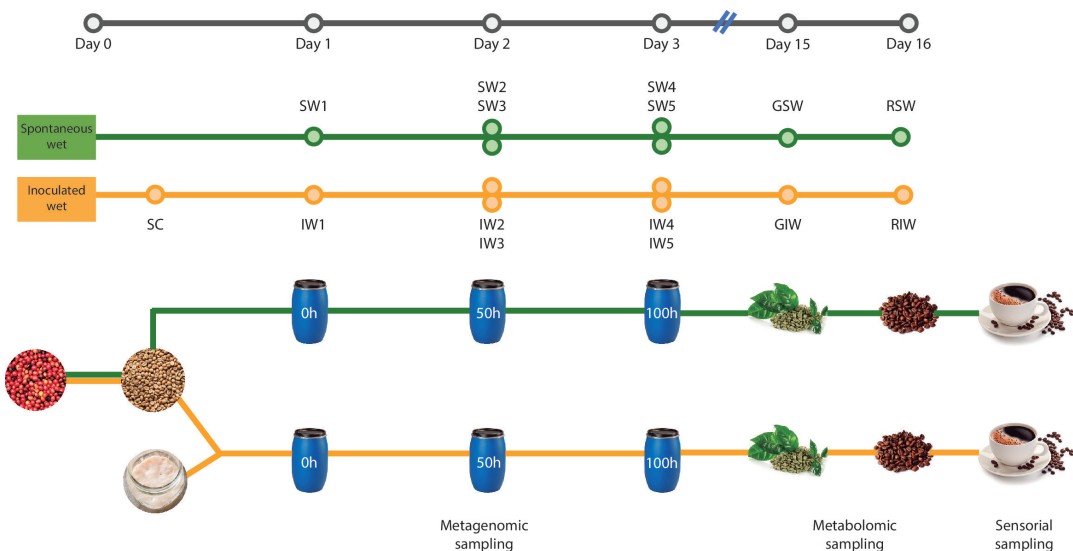

**FIG 1** Experimental setup of the study of two coffee-processing processes. The green line depicts the SW and the orange line the IW. Sample SW1 refers to the liquid fraction of the fermentation process collected at the start (0 h). Samples SW2 and SW3 correspond to the liquid fraction collected at 50 h (midpoint of the fermentation) from the top and middle of the barrel, respectively. SW4 and SW5 correspond to the liquid fraction collected at 100 h (endpoint of the fermentation) from the top and middle of the barrel, respectively. Sample SC refers to 50 mL of the starter culture that was taken before the start of the fermentation. Samples IW1–IW5 correspond to the liquid fraction of the fermentation in equivalent times and sections of the barrel as in SW. Samples GSW and GIW correspond to the dehulled, green coffee beans from the spontaneous and inoculated fermentations, respectively, and samples RSW and RIW correspond to the roasted beans from the spontaneous and inoculated fermentations, respectively.

they were kept at −80°C until further analyses. The sampling scheme and sampling times are depicted in Fig. 1.

## Roasting and sensory evaluation

The green coffee beans from the SW and IW fermentation processing (150 g) were roasted according to roasting protocols established by the producers. Figure S1 shows the roasting curves corresponding to the SW and IW coffee. Cupping was done by three certified Q-grader panelists. The beverages were prepared at a ratio of 8.5 g of roasted and ground beans per 150 mL of water. They were brewed at 100°C and cupped at 71°C in espresso glass cups one at a time, following the standard protocol established by the Specialty Coffee Association (26). A quantitative descriptive analysis was applied to measure the intensity of the sensory attributes given by the panelists, covering aroma, flavor, acidity, and body, among others. A one-way analysis of variance on ranks (Kruskal-Wallis test) was used to evaluate the statistical significance of the results, and a Wilcoxon test was performed to calculate pairwise comparisons between group levels ($P < 0.05$). An unpaired two-sample $t$-test was conducted for the determination of differences in attribute scores for all three cuppers between both fermentations using the statistical analysis software R version 4.3.0.

## Untargeted metabolomic analysis

An untargeted metabolomics analysis was used to determine metabolite variation in both types of fermentations (SW and IW) for the resulting green and roasted coffee beans following fermentation and roasting using Reverse Phase Liquid Chromatography system coupled to a Quadrupole-Time-Of-Flight mass selective detector (RP-LC-MS-QTOF), Hydrophilic Interaction Chromatography system coupled to a Q-TOF Mass Spectrometer (HILIC-LC-MS-QTOF), and Gas Chromatography coupled to a Q-TOF Mass Spectrometer (GC-MS-QTOF). Detailed information for each platform and preparation of samples and quality control (QC) samples can be found in the "Methods" section of the supplemental material.

Agilent MassHunter Profinder 10.0 was used to evaluate the data from RP-LC-MS-QTOF analyses, while Agilent MassHunter Unknowns Analysis B.10.00 and Agilent MassProfiler Professional were used for data deconvolution and alignment in GC-MS-QTOF. The data were filtered for reproducibility with a co-efficient of variation (CV) of the area in the QC samples. A CV > 20% for RP-LC-MS-QTOF, and a CV > 30% for GC-MS-QTOF. On the other hand, only data with 100% absence or presence in at least one group were retained. The study used unsupervised principal component analysis (PCA) to verify analytical platforms' reproducibility and sample dispersal. Comparisons between green and roasted coffee beans were evaluated using univariate (UVA) and multivariate statistical analysis (MVA). Supervised partial least squares-discriminant analysis (PLS-DA) models were used to select molecular characteristics for group separation, meeting UVA ($P < 0.05$) or MVA (Variance Important in Projection > 1 with Jack-Knife confidence interval not including 0) for GC-MS-QTOF, while UVA and MVA with the same conditions were used for RP-LC-MS-QTOF.

Metabolite identification for RP-LC-MS-QTOF was performed using databases like the Human Metabolome Database, Kyoto Encyclopedia of Genes and Genomes, MassBank, Lipid MAPS, and METLIN. Isotopic mass, isotopic distribution, adduct formation, retention time, MS/MS spectra, and molecular formulas generated and confirmed using Agilent MassHunter Qualitative Software 10 were also considered. The confidence levels for the annotation were reported according to the guidelines presented in previous studies (27). GC-MS-QTOF metabolite identification was performed using Fiehn libraries, PCDL Manager B.08.00, and NIST. Databases like the Human Metabolome Database, FooDB, and The Good Scents Company were used to identify organoleptic characteristics and biological origin. Signals with a significant statistical variation were selected for annotation. For more details about the methodology used, view the "Methods" section in the supplemental material.

Heatmaps showing the relative abundance of annotated metabolites for green and roasted coffee beans were generated using the ggplot2 package (28) in the software R version 4.3.0. The relative abundances of each metabolite across samples were normalized using the $z$-score method. Metabolites were grouped by either the sensorial profile to which they are associated, or if none could be found, the origin of said metabolite (bacterial, fungal, coffee plant, or other).

## DNA extraction and metagenomic analysis

Total DNA was extracted from the liquid fraction of fermenting coffee beans at three different time points (0, 50, and 100 h) using the DNeasy PowerSoil Pro DNA Isolation Kit (QIAGEN, Hilden, Germany) according to the manufacturer's instructions, with minor modifications. Before the extraction, 600 µL of the liquid fraction of the fermenting coffee beans was washed with sterile phosphate-buffered saline 2×. The resulting pellet was processed for the extraction of the bacterial and fungal DNA. The genomic DNA was quantified with the Nanodrop 2000 spectrophotometer (Thermo Fisher Scientific, Waltham, MA, USA). The DNA was exported under permit 02495 from the National Authority of Environmental Licenses (ANLA, Colombia). PCR amplification, library preparation, and sequencing were performed as a service at the DNA Sequencing Innovation Laboratory at the Edison Family Center for Genome Sciences, Washington University School of Medicine. The specific V4 region of the bacterial 16S gene was amplified using the F515 and R806 primers, and the fungal ITS1 region was amplified using the primers ITS1f and ITS2. Bacterial amplicons were pooled and purified with 0.6× Agencourt Ampure XP beads (Beckman-Coulter) prior to sequencing using the 2 × 250 bp protocol on the Illumina MiSeq platform (29). Fungal amplicons were processed following the ITS protocol from the Earth Microbiome Project (30). All samples were processed on the same day with one batch of reagents by the same technician and sequenced on a single MiSeq run to minimize batch effects. Four Fastq files were obtained for each sample, corresponding to the forward and reverse reads for both the bacterial and fungal amplicons.

Sequencing data were received demultiplexed from the DNA Sequencing Innovation Laboratory corresponding to the forward and reverse reads for each sample for both the bacterial and fungal samples. The QIIME2 software (version 2021.11.0) was used for all the initial bioinformatic processing, including QC and data trimming, as well as for the inference of amplicon sequence variants (ASVs) using the DADA2 plugin by denoising, read merging, and chimera removal. This software was also used for taxonomic, alpha, and beta diversity analyses using the q2-taxa and q2-diversity plugins, respectively. Taxonomy was assigned with the SILVA 138 database for the bacterial ASVs and with the UNITE database (version 8) for the fungal ASVs using trained Naive Bayes classifiers.

The microbial relative abundances for both the bacterial and the fungal samples were statistically analyzed using software R version 4.3.0, implementing the qiime2r package (31) as QIIME2 artifacts. Additionally, the ggplot2 package (28) was used to construct a principal coordinate analysis (PCoA) plot based on Bray-Curtis dissimilarities, and an analysis of similarities (ANOSIM) test was performed to determine statistically significant differences in the clustering of samples. The sequence relative abundance data were used to construct a Spearman's rank-order correlation matrix to evaluate the dependence among the taxa. Figures were generated using the corrplot package (32). An unpaired two-sample $t$-test was conducted for the determination of differences in microbial abundances between both fermentation processes. A probability level of 0.05 was considered as significant for all statistical analyses.

## RESULTS

### Inoculated fermentation improved coffee quality

The scoring for a cup of coffee is based on a matrix evaluating variables such as flavor, aroma, aftertaste, and body. Scores from three certified Q-grader panelists for each attribute can be found in Fig. 2; Table S1.

Among the evaluated variables, uniformity and cleanness differed significantly between coffee prepared from beans processed via spontaneous (SW) and inoculated (IW) fermentations (Fig. 2). The SW coffee was characterized by phenolic notes, negatively affecting the uniformity and cleanness scores. Other organoleptic descriptors further distinguish the SW from IW coffee, with the SW being described as less sweet, more acidic, and having chocolate-like, caramel, coconut, and peppery notes. In contrast, the IW coffee was perceived as sweeter and less acidic, with a cleaner cup (free of taints and defects), featuring notes of pineapple, eucalyptus, white chocolate, and wine, along with floral flavors and aromas.

### Different metabolomic profiles for inoculated and spontaneous fermentations

The PCA model in Fig. S2 includes all samples and QC from each analysis. Clustering of the QC samples confirmed that observed variations were sample-specific, not analytical artifacts.

PLS-DA comparing green coffee from both groups (SW and IW) revealed a distinct separation between the green spontaneous (GSW) and the green inoculated (GIW) coffee, indicating different metabolic composition (Fig. 3). High coefficient determination ($R^2 > 0.9$) and the prediction capacity ($Q^2 > 0.6$) across all analytical platforms indicate robust models.

The complete list of significantly differential compounds between spontaneous and inoculated processing for both green and roasted coffee beans can be found in Table S2 and S3, respectively. Between GSW and GIW, 59 compounds were identified and between RSW and RIW beans, 77.

### Metabolites associated with organoleptic profiles differ by fermentation type

The normalized relative abundances of identified metabolites associated with specific organoleptic profiles are shown in Fig. 4. In agreement with the coffee cuppers who

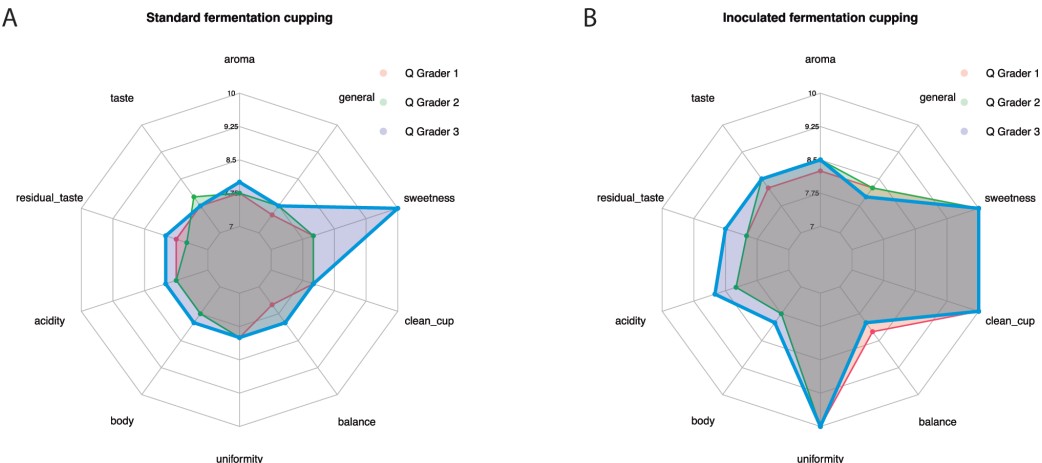

**FIG 2** Sensorial analysis of coffees brewed from the (**A**) SW and (**B**) IW coffee processes by the three Q-graders. Scores for each feature are evaluated by each grader with a score from 1 to 10. Larger shaded areas correspond to higher overall scores.

reported higher sweetness in the IW, GIW coffee had higher levels of metabolites linked to sweetness (Fig. 4A) when compared to GSW. We detected an increase of compounds like alanine, fructose, mannitol, sorbitol, and sucrose which have been reported as being responsible for sweet flavor found in foods and beverages (33). Roasted coffee beans exhibited similar trends (Fig. 4B). RIW beans had higher abundances of metabolites associated with sweetness and fatty/buttery notes and of metabolites linked to sweetness such as methyl phenylacetate, phenylethyl formate, fructose, maltose, lactose, galactose, vanillic acid, mannitol, sorbitol, and 2-ketobutyric acid.

Regarding the GSW beans, these showed significantly higher abundances of metabolites associated with caramellike, fruity, bitter, and acidic notes ($P < 0.05$). Moreover, cinnamic acids such as feruloylquinic acid and feruloyl-quinolactone were more abundant for these coffee beans, which could possibly be responsible for the higher levels of acidity for this coffee (34). Similarly, RSW beans had more metabolites associated with fruity and caramellike notes, with higher levels of butyrolactone, ethyl pyridine, and pyruvic acid, all associated with caramellike and brown sugar flavors and aromas. Methoxymethyl furan can possibly explain the roasted notes, while isopropyl formate, furfuryl acetate, and ethyl vanillin the fruity notes. The SW coffee was described as having notes of caramel, which could be explained by the higher abundance of 2-ketobutyric acid, acetol, furancarboxaldehyde, ethyl pyridine, ethyl vanillin, and/or pyruvic acid. Furthermore, considering the described sensorial profiles of both coffees, the SW fermentation was characterized by a tangible defect, a phenolic taste, perceived by the three certified cuppers.

Finally, GIW displayed higher abundances of fungal metabolites, including fumonisins and myo-inositol, while GSW had higher abundances of metabolites produced by bacteria, like phenyllactic acid and arabinose. Other compounds frequently found in coffee, like cafestol, were found with variable abundances among fermentation types and no association with particular organoleptic attributes. RIW beans also had higher levels of malic acid, oxalic acid, pyruvic acid, quinic acid, citric acid, coumaroylquinic acid, and 3,4-dicaffeoylquinic acid, contributing to a desirable sensorial profile (34, 35).

## Inoculation with a starter culture is not the most important factor in microbial community composition

Beta diversity analysis derived from the amplicon sequencing of bacterial and fungal marker sequences (Fig. 5) showed that time of fermentation significantly affects microbial structure more than inoculation. For bacterial and fungal analysis, the initial (0 h) community shows the greatest dissimilarity, which was to be expected given the

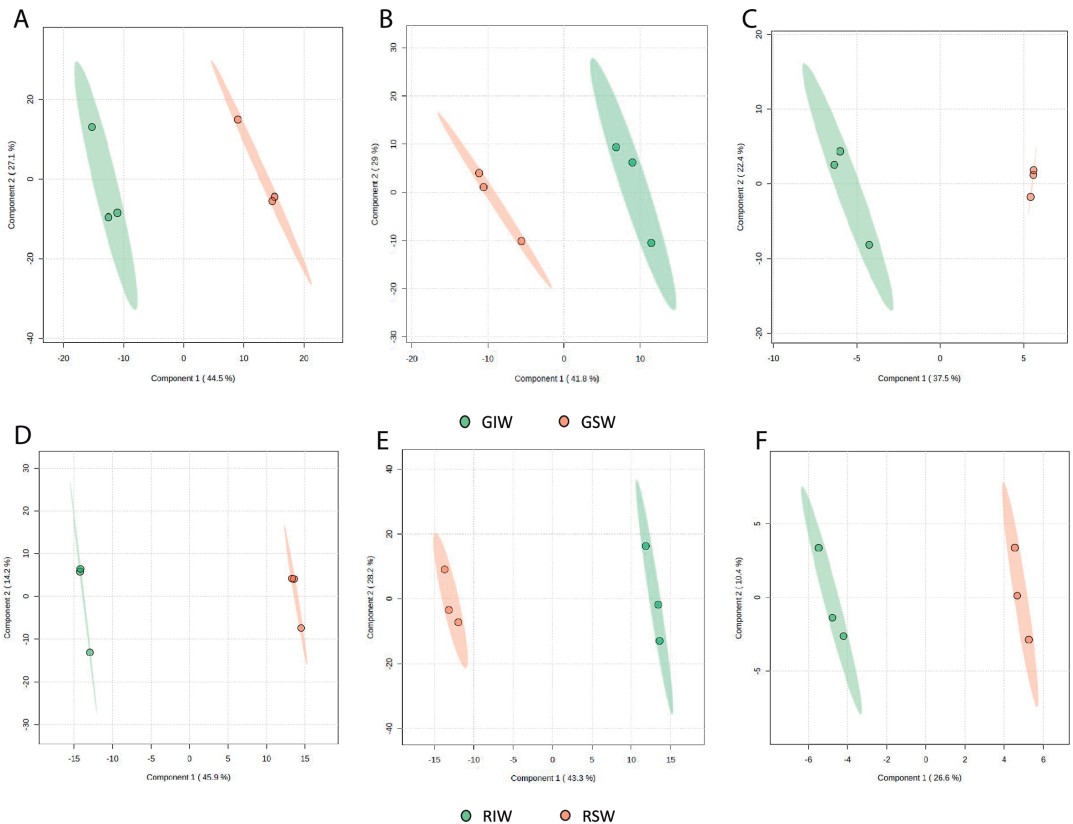

**FIG 3** PLS-DA plots for the comparison between both green and roasted coffee bean groups and the different non-targeted metabolomics platforms used. Panels A–C are for green coffee beans, while panels D–F are for roasted coffee beans. Panels A and D used RP-LC-MS, B and E used HILIC–LC-MS, and C and F used GC-MS. (**A**) RP-LC-MS (+): $R^2X_{(cum)}$: 0.96852, $R^2Y_{(cum)}$: 0.99579, $Q^2_{(cum)}$: 0.81098; (**B**) HILIC-LC-MS: $R^2X_{(cum)}$: 0.94402, $R^2Y_{(cum)}$: 0.99195, $Q^2_{(cum)}$: 0.71001; and (**C**) GC-MS: $R^2X_{(cum)}$: 0.98645, $R^2Y_{(cum)}$: 0.99731, $Q^2_{(cum)}$: 0.68453. (**D**) RP-LC-MS (+): $R^2X_{(cum)}$: 0.99841, $R^2Y_{(cum)}$: 0.9994, $Q^2_{(cum)}$: 0.83429; (**E**) HILIC-LC-MS: $R^2X_{(cum)}$: 0.99659, $R^2Y_{(cum)}$: 0.99937, $Q^2_{(cum)}$: 0.87468; and (**F**) GC-MS: $R^2X_{(cum)}$: 0.99223, $R^2Y_{(cum)}$: 0.99887, $Q^2_{(cum)}$: 0.43977. GSW: green coffee processed with the spontaneous fermentation method. GIW: green coffee processed with the inoculated fermentation method. RSW: roasted coffee processed with the spontaneous fermentation method. RIW: roasted coffee processed with the inoculated fermentation method.

inoculation with the starter culture. Then, for bacterial communities, time (50 h vs 100 h) was a more important driver of variation than fermentation type (Fig. 5A), while for fungal communities, the opposite was observed (Fig. 5B).

## Inoculated fermentation had higher bacterial and lower fungal diversity

Rarefaction analyses indicated sufficient sequencing depth for bacterial and fungal diversity studies (Fig. S3A and B, respectively). Bacterial diversity (Shannon index) and richness (observed index) were higher than fungal in both processes, with the greatest bacterial diversity and richness being seen in the initial hour of the spontaneous fermentation (SW1).

However, some variations were observed between both types of fermentation (Fig. 6). SW fermentation had a greater fungal diversity and richness than the IW fermentation, with no significant differences in bacterial diversity or richness ($P = 0.21$ and $= 0.16$, respectively). A lower value to the Shannon index in the IW fermentation could indicate the dominance of the starter culture used in this type of process (Fig. 6B). High microbial diversity at hour zero of fermentation indicates microbial diversity originating from the coffee fruit itself, with diversity decreasing as fermentation progresses due to microorganism specialization or inhibition by stable microorganisms after hour 50. This selection of microorganisms is likely responsible for the observed changes in chemical composition between spontaneous and inoculated green and roasted coffee beans.

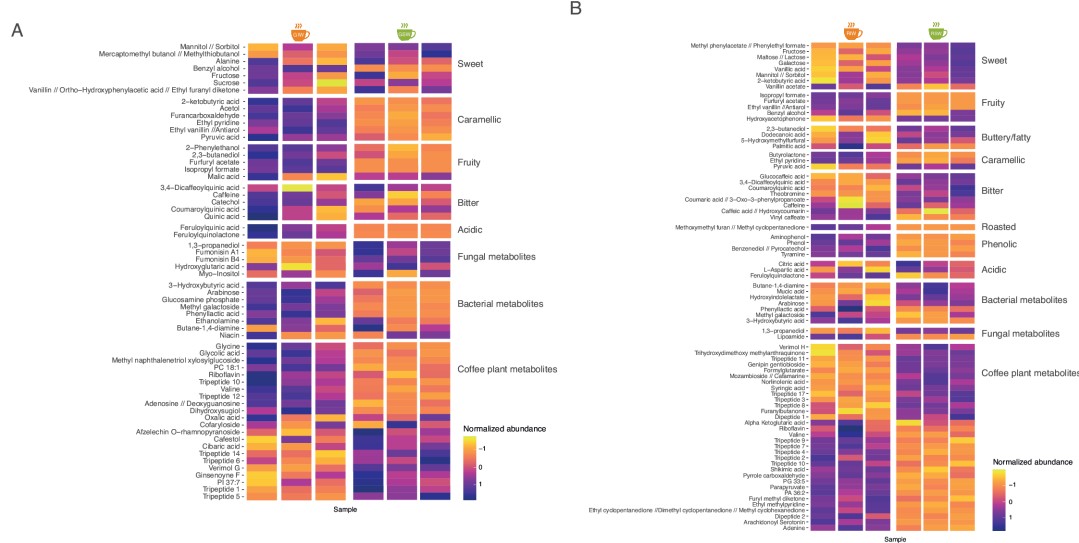

**FIG 4** Heatmaps displaying the normalized relative abundance for the (**A**) green coffee and (**B**) roasted coffee beans from both types of fermentations. Lower *z*-scores are represented by yellow and signify higher relative abundances. Higher *z*-scores are represented by purple and signify lower relative abundances. Metabolites are categorized either on the sensory profile they are associated with or, if no sensory attributes can be attributed to them, the origin of the metabolite (bacteria, fungi, or coffee plant).

## Inoculated fermentation is dominated by LAB

Relative abundances of microbial taxa are shown in Fig. 7. *Lactobacillus* and *Leuconostoc* dominated the bacterial communities, while *Kazachstania humilis* and a member of Saccharomycetales characterized the fungal communities. At hour zero of IW, *Lactobacillus*, *Lactobacillus brevis*, and *Lactobacillus fermentum* were significantly more abundant, along with the added yeast. At hour 50, *Lactobacillus* and *K. humilis* significantly increased for both fermentations.

## Co-absence of *Lactobacillus* and *Kazachstania* with most taxa

The Spearman's rank correlation plot (Fig. 8) revealed strong negative correlations between four taxa, two species of *Lactobacillus* (*L. brevis* and *L. fermentum*) and two species of *Kazachstania* (*K. humilis* and *Kazachstania exigua*), against several other reported taxa during the final hour of fermentation, while displaying weak positive correlations with the added yeast and yeasts of the genus *Pichia*. This pattern likely originates from the enrichment and dominance of *Lactobacillus* and *Kazachstania* taxa toward the latter stages of fermentation. As previously mentioned, microbial and fungal diversity markedly decreases at hours 50 and 100 of fermentation (Fig. 6A and B), suggesting preferential selection of *Lactobacillus* and *Kazachstania* under the prevailing environmental conditions within the fermentation tanks across both inoculated and spontaneous processes.

## DISCUSSION

To our knowledge, this study is the first to investigate the co-fermentation of coffee with a starter culture, a consortium of selected bacterial and fungal microorganisms combined with other sources of microbial diversity, to enhance the organoleptic characteristics of a commercially available coffee. Our systematic approach, encompassing the monitoring of both microbial communities and metabolomic profiles during two different coffee fermentations, demonstrated that the addition of starter cultures to coffee fermentations significantly alters both the metabolic and microbial composition of the fermentation process, which could be associated with differences in quality of the

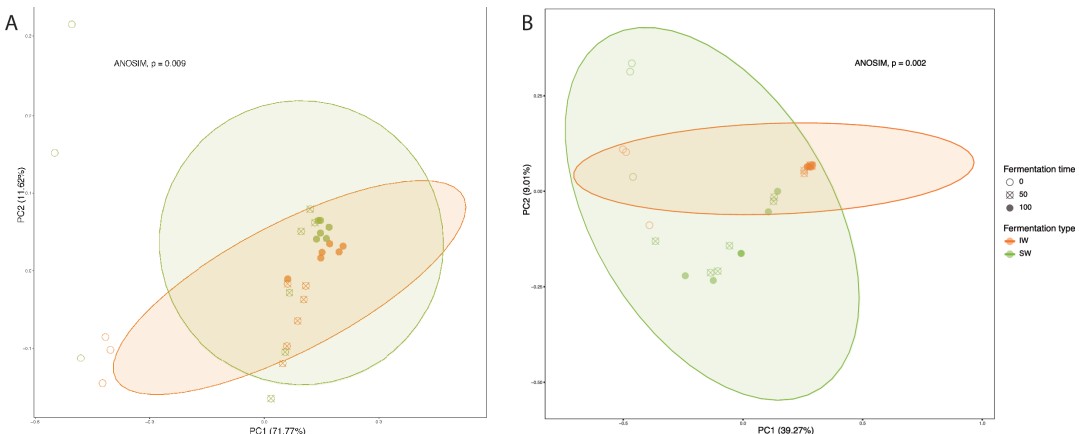

**FIG 5** PCoA plots based on Bray-Curtis dissimilarities of: (**A**) bacterial and (**B**) fungal communities of the different fermentation processes sampled and analyzed, inoculated (IW) and spontaneous (SW).

resulting beverage. This improvement was supported by higher scores assigned to the inoculated coffee by certified coffee cuppers.

Both the SW and IW were characterized by similar abundances of bacterial taxa, with LAB and *Kazachstania* yeast being the most abundant throughout the entire process. LAB have been previously reported at high abundance in wet coffee fermentations (10, 36). Particularly for Colombian coffee, *Leuconostoc* and *Lactobacillus* have ubiquitously been reported as the most important and predominant genera during the fermentation process of high-quality coffee (36–39). All LAB species found are heterofermentative, capable of producing a vast range of metabolites, including lactic acid, acetic acid, $CO_2$, and ethanol. *L. fermentum* has been reported to additionally produce 2-methyl-butanol and ethyl butanoate, while *L. brevis* produces 2-methyl-butanol and phenylacetaldehyde, and species of *Leuconostoc* ethyl acetate, ethyl butanoate, and ethyl hexanoate (36), all associated with desirable sensorial descriptions, such as fruity and floral notes. LAB also suppress mycotoxigenic fungal growth and inhibit production of mycotoxins (40), supporting their importance in coffee fermentations.

To our knowledge, *Kazachstania* has only been reported three times before in a coffee fermentation (39, 41, 42). Previously, it had been reported in coffee fermentation wastewater (43) and in sourdough fermentations (44). Studies have linked *Kazachstania* species to the production of flavor compounds like isobutyric and hexanoic acids in grape juice fermentations (45), which have been associated with sweet and fruity flavors, respectively. It is unlikely, however, that this yeast solely originates from the wheat flour used in the starter culture to inoculate the coffee fermentation, since this genus is found in both of the studied fermentation processes. It is more likely, then, that *Kazachstania* is part of a broader microbial community that naturally colonizes coffee beans during fermentation. Its presence in both inoculated and spontaneous fermentations reflects the resilience of the fermentation regardless of the different processes.

It is important to note that both spontaneous (SW) and inoculated (IW) fermentations exhibited broadly similar microbial compositions across the 100 hsampling window. However, the IW fermentation showed notably faster stabilization, particularly within the fungal community: by 50 h, the composition had reached a plateau, and no new taxa emerged by 100 h, suggesting that the process is largely driven by shifts in the relative abundances of pre-existingmicroorganisms, rather than the introduction of additional species. This pattern aligns with findings from other controlled fermentations, where inoculation with a defined starter cultureaccelerates microbiota succession and narrows microbial diversity earlier in the process. For instance, Polonía-Rivera et al. (46) showed that starter-mediated fermentation not only enhanced sensory quality but also shortened processing times by driving rapid microbial dominance and efficient mucilage

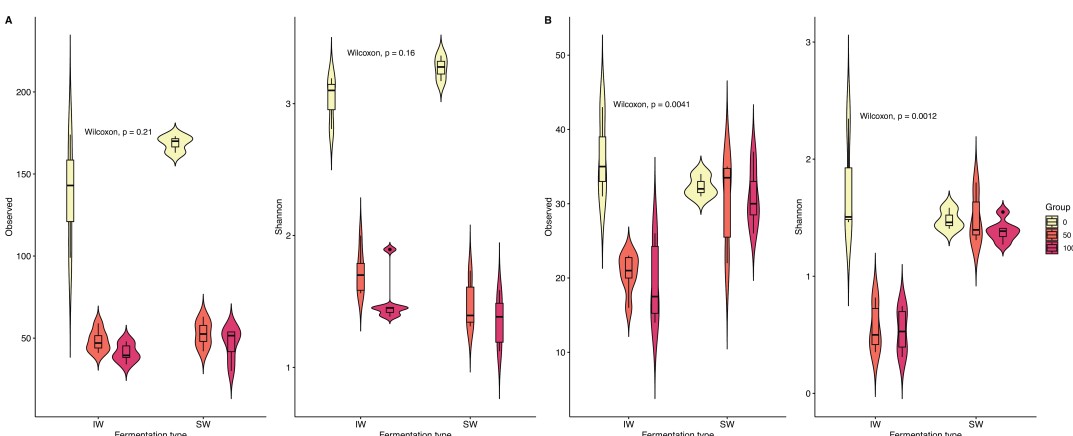

**FIG 6** Alpha diversity indices visualized as box and kernel density plots for (**A**) bacterial and (**B**) fungal communities. Horizontal bars in the boxplots indicate the medians of each group. The Observed and Shannon indices are provided. The two groups are the inoculated (IW) and spontaneous (SW) fermentations, grouped by time of fermentation (0 h, yellow; 5 h, orange; 100 h, red).

degradation. Similarly, controlled fermentations in other studies have shown that the early establishment of selected microbes can streamline metabolite production and stabilize the microbial ecosystem more quickly than spontaneous fermentations (47).

Although we did not determine the exact microbial composition of the used starter culture, beyond the presence of the two commercial yeasts, our results suggest that its microorganisms did not directly colonize the fermentation, as both SW and IW fermentations exhibited similar microbial diversity and taxa. Instead, the differences observed appear to be related to shifts in relative abundances of the resident community, likely driven by the additional nutrients introduced through the inoculum. This nutrient enrichment may have facilitated faster microbial activity and more efficient metabolite production, ultimately leading to a higher-quality sensory outcome. Similar effects have been described in other food fermentations, where supplementation with simple sugars or nutrient-rich substrates enhanced microbial metabolism and improved flavor compound formation (9, 46). In future work, it will be essential to disentangle the contributions of each inoculum component, whether microbial, nutritional, or synergistic, to better understand their role in shaping coffee fermentation outcomes and to inform the development of standardized, commercially viable starter cultures.

Future studies should aim to elucidate the mechanisms of compound permeability during fermentation. Specifically, it will be important to measure metabolomic changes not only in the fermentation liquid but also in the coffee grain before and after processing. This would allow us to determine the bidirectional exchange of metabolites, demonstrating which compounds from the liquid phase penetrate into the grain and which intracellular metabolites are released into the surrounding medium. Furthermore, applying microscopy-based approaches to visualize structural changes in the coffee endosperm could provide evidence of increased permeability and reveal whether inoculated fermentations accelerate or enhance compound diffusion. Such analyses would clarify the extent to which metabolite transfer between the fermentation medium and the coffee bean explains the chemical and sensory differences observed in the final product.

Although green coffee beans experience complex chemical transformations during roasting (48), metabolic profiles of the green coffee are strongly correlated with the quality of the coffee beverage (49, 50). Meanwhile, the Maillard reaction and other complex chemical transformations during roasting (51–53) alter the metabolic profiles of the coffee beans, contributing to flavor and aroma changes (50), meaning the metabolites found for roasted coffee are also directly correlated to the organoleptic profiles of the brewed coffees.

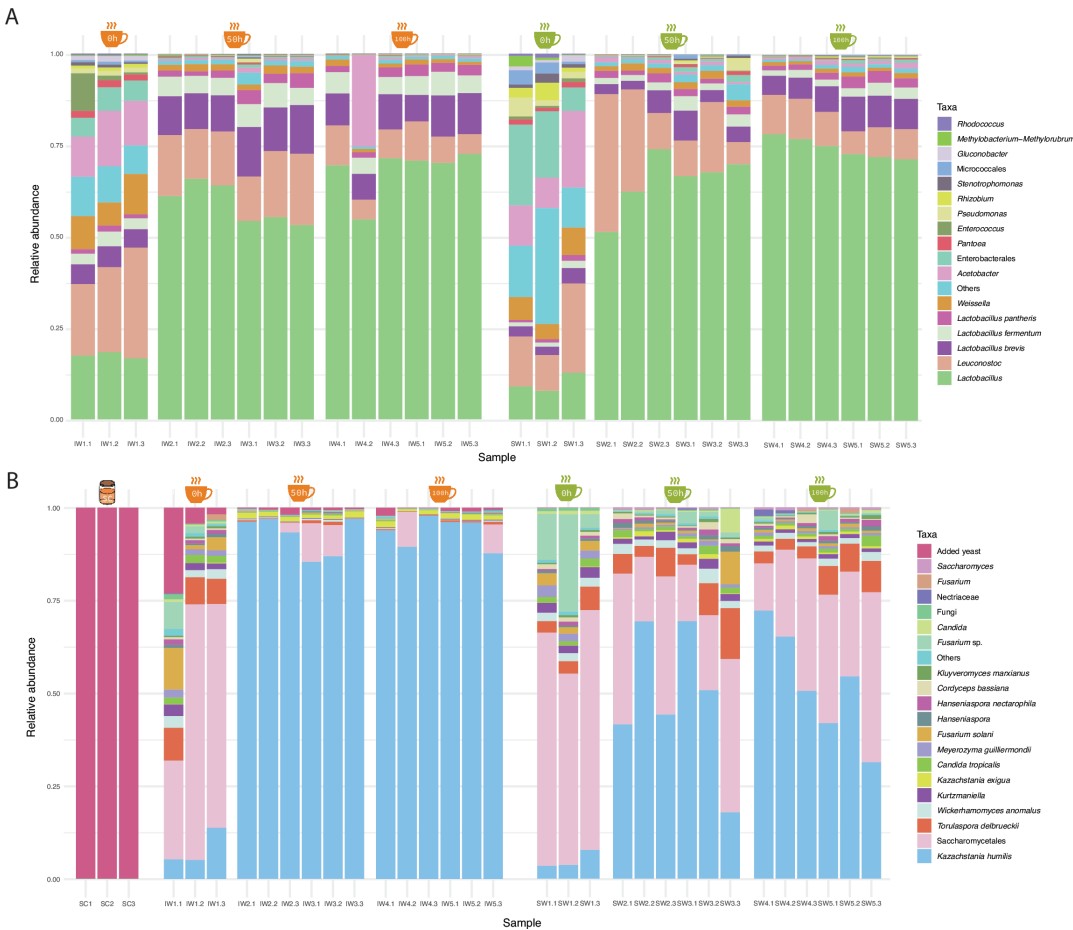

**FIG 7** Relative abundances of bacterial (**A**) and fungal (**B**) ASVs present in the selected samples for the spontaneous (SW) and inoculated (IW) fermentation processes, as well as the fungal starter culture (SC) used for the inoculation. Detected ASVs below 1% in at least one sample are indicated as "others."

The presence of defects is crucial in determining the quality of the coffee. Phenolic compounds are usually formed due to improper processing conditions that result in "sour" and defective beans (54–56). We identified the presence of two phenolic compounds, aminophenol and phenol, in significantly higher abundance in the samples from the RSW beans and could explain the phenolic taste, reminiscent of plastic or rubber. Additional compounds that have also been associated with phenolic organoleptic qualities, benzenediol, and tyramine were also found in the RSW coffee beans. To the best of our knowledge, this is the first time that the association between the phenolic defect and chemical composition has been reported. This is crucial because phenol notes can only be detected during the cupping stage, and they cannot be linked to any physical defects in the beans. The inoculation process may help control the emergence of cup defects like phenol by influencing the microbiota involved in fermentation.

Due to the brief duration of fermentation, discerning whether the enriched microorganisms in the latter stages of both fermentation processes are predictive of the identified metabolic traits in green and roasted coffee beans, as well as the sensory profiles of brewed and cupped coffees, poses a challenge. To gain deeper insights into this, untargeted metabolomic profiling spanning the entire fermentation timeline, from hour zero to 100, would be necessary. Such an approach would enable the identification of the onset of various reported metabolites during fermentation. While this type of analysis would be ideal, our available data limit us from detecting a direct correlation between specific microorganisms and individual metabolites. Nonetheless,

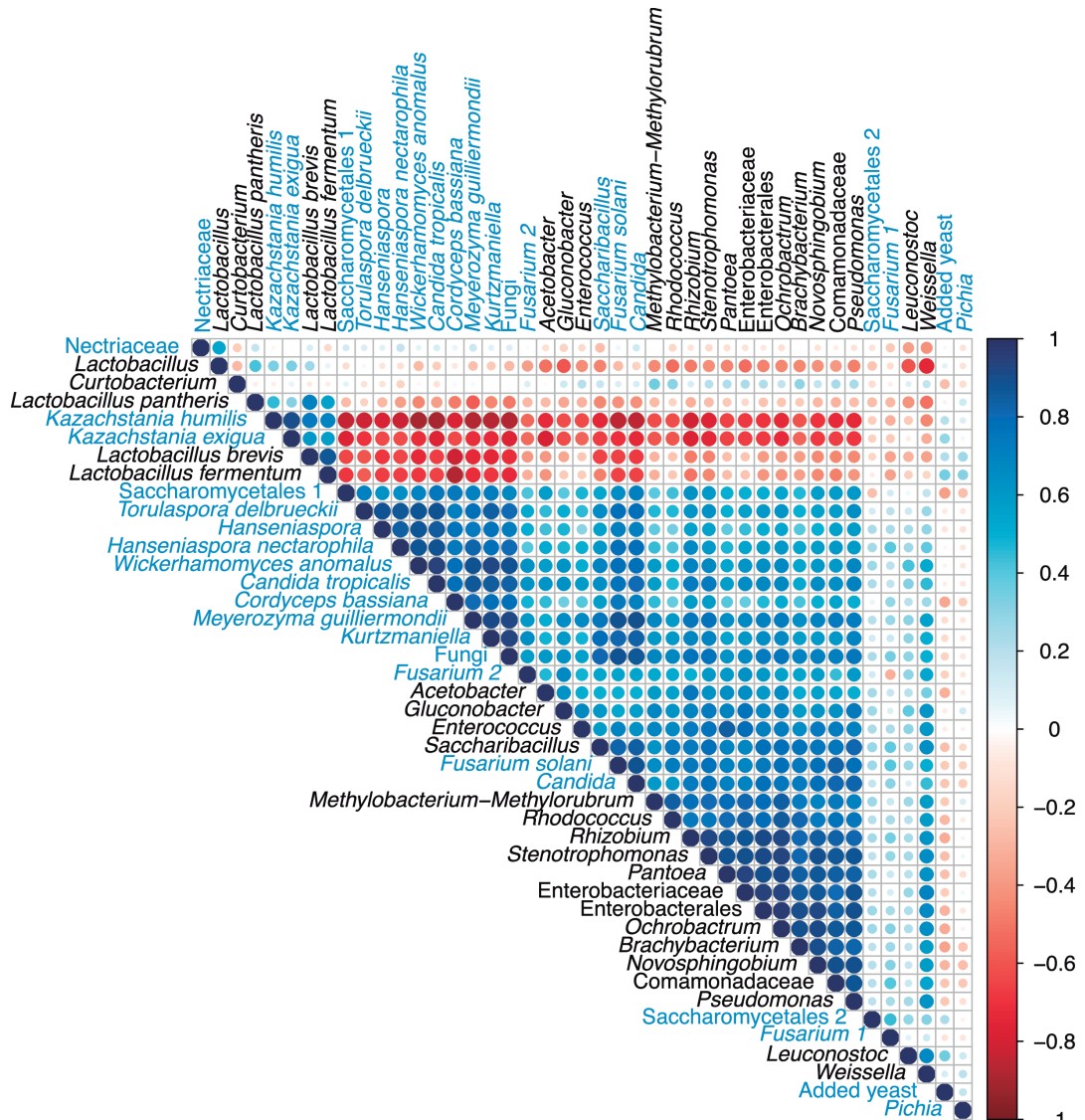

**FIG 8** Spearman's rank correlation plot of microbial taxa identified (>1% abundance in at least one sample) from both fermentation processes. Taxa names in black correspond to bacteria and in blue to fungi. Circle sizes are proportional to the strength of the correlation. The colors of the scale bar indicate the nature of the correlation; dark blue indicates a perfect positive correlation (co-presence), and dark red indicates a perfect negative correlation (co-absence).

existing literature provides clear examples of correlation between microorganisms and metabolites.

Fungal metabolites such as hydroxyglutaric acid, 1,3-propanediol, and myo-inositol, which are all known to be produced by the strain of yeast used in the starter culture, were all identified in the GIW samples. Regarding bacterial metabolites, their abundance varied depending on the compound. Some were more abundant in the GSW, like phenyllactic acid, methyl galactoside (57), and glycolic acid (58), whereas others had higher abundances in GIW beans, like mucic acid, mannitol, sorbitol, and L-aspartic acid. Since neither the diversity nor richness of bacterial taxa varied significantly between fermentations, it is possible that the microbial context in which these microorganisms are found contributes to differential metabolomic profiles. This highlights the importance of having complete and annotated metabolomes for bacterial and fungal species. Other microbial metabolites, like acetic acid and glycerol (59, 60), were not found with the metabolomics approach used in this study. However, their presence could be inferred

given the abundance of bacteria like *Acetobacter*, *Lactobacillus*, and *Gluconobacter* during the fermentation process.

Some of the metabolites identified as significantly abundant for the GIW coffee beans have been reported to be associated with the previously described microorganisms. Compounds such as glucosamine phosphate and mercaptomethyl butanol, known to be produced or metabolized by bacteria and fungi, were found to be significantly abundant in the GIW coffee. Glucosamine phosphate is an organic compound known to be produced by both bacteria and fungi (61). Moreover, mercaptomethyl butanol, an organosulfur compound characteristic of coffee which contributes to its distinct roasted aroma and sweetness, has been reported to be produced during fermentation by bacteria. In particular, LAB are known to produce butanol through microbial fermentation (62).

In addition to the metabolites identified in the GIW beans, the GSW beans were also characterized by a diverse array of compounds, reflecting the complex metabolic interactions within the microbial community aligning with its higher fungal and slightly higher bacterial diversity. Notable constituents include 1,3-propanediol, a known fermentation product of certain bacteria and yeasts, including LAB and *S. cerevisiae* (63–65) and 2-phenylethanol, which is often associated with yeast metabolism and contributes to the aromatic complexity of fermented coffee beans (66). Pyruvic acid, a key intermediate in glycolysis, may be produced by various microbial taxa and can be converted into other products crucial for the determination of flavor and aroma in coffee, such as succinic acid, lactic acid, and acetic acid through glyceropyruvic fermentation (67). The presence of glycolic acid and hydroxyglutaric acid suggests the involvement of metabolic pathways like ethylene glycol oxidation in bacteria (58) and D-lactate dehydrogenase metabolism in yeast (68). The presence of arabinose, a common sugar, and methyl galactoside, a derivative of galactose, may indicate the breakdown of complex carbohydrates present in the coffee beans by microbial enzymatic activity.

## Conclusions

Our study, comprising monitoring of both microbial communities and metabolomic profiles during two different coffee fermentations, demonstrated that the inoculated fermentation resulted in a better cup quality for the brewed coffee. This inoculation process with a starter culture significantly improved the metabolite composition, contributing to the perception of flavor and aroma notes desired by consumers. Furthermore, the inoculated fermentation was characterized by high abundances of LAB and yeasts of the genus *Kazachstania*, microorganisms associated with the production of desirable metabolites during coffee fermentation such as lactic, acetic, isobutyric, and hexanoic acids. Further studies should focus on thoroughly understanding the influence of microbial starter cultures on coffee quality and the interactions between microorganisms and metabolites. In addition, more research is needed on the use of starter cultures to inoculate wet fermentation processes and produce new and innovative flavor profiles in specialty coffee.

## ACKNOWLEDGMENTS

We gratefully acknowledge Café Fino S.A.S. for their partial financial support of this research project and for providing the coffee samples used in our study. We also acknowledge the Max Planck Tandem Group in Computational Biology at Universidad de los Andes for funding of M.A.M.-R. and some of the DNA extractions. Special thanks go to the IT Services Department at the Universidad de Los Andes for providing high-performance computing services. We acknowledge the instruments and scientific and technical assistance of the Metabolomics Core Facility – MetCore at the Universidad de Los Andes, a facility supported by the vice-presidency for research. Additional thanks go to Julio Madrid and Andrés Quiceno for their roles in sampling and processing the coffee samples. Finally, we extend our gratitude to Adriana Bernal and her research group

LIMMA at Universidad de los Andes for providing lab space for DNA extractions and PCR reactions.

This research did not receive any specific grant from funding agencies in the public or not-for-profit sectors. The company Café Fino S.A.S. provided funding for sample collection, shipping, and metabolomic analysis. All further funding came from the Max Planck Tandem group and the Metabolomics Core Facility – MetCore at the Universidad de Los Andes.

M.A.M.-R.: conceptualization; methodology; formal analysis; investigation; writing—original draft; visualization; and funding acquisition. A.M.L.-I.: methodology, formal analysis, investigation, and writing—original draft. A.E.P.-M.: conceptualization, methodology, and writing—review and editing. M.P.C.: methodology, resources, writing—review and editing, and supervision. A.R.: conceptualization; methodology; validation; investigation; resources; writing—review and editing; supervision; and funding acquisition.

## AUTHOR AFFILIATIONS

[1]Group in Computational Biology and Microbial Ecology, Department of Biological Sciences, Faculty of Science, Universidad de los Andes, Bogotá, Colombia
[2]Max Planck Tandem Group in Computational Biology, Faculty of Science, Universidad de los Andes, Bogotá, Colombia
[3]MetCore - Metabolomics Core Facility, Vice-Presidency for Research, Universidad de los Andes, Bogotá, Colombia
[4]National Coffee Research Center, Cenicafé, Manizales, Colombia

## PRESENT ADDRESS

Maria A. Madrid-Restrepo, KU Leuven, Leuven, Belgium

## AUTHOR ORCIDs

Maria A. Madrid-Restrepo http://orcid.org/0000-0003-2383-6161
Ana M. León-Inga http://orcid.org/0009-0001-7218-8693
Aida Esther Peñuela-Martínez http://orcid.org/0000-0003-4454-9778
Mónica P. Cala http://orcid.org/0000-0002-8198-726X
Alejandro Reyes http://orcid.org/0000-0003-2907-3265

## AUTHOR CONTRIBUTIONS

Maria A. Madrid-Restrepo, Conceptualization, Formal analysis, Funding acquisition, Investigation, Methodology, Visualization, Writing – original draft | Ana M. León-Inga, Formal analysis, Investigation, Methodology, Visualization, Writing – original draft | Aida Esther Peñuela-Martínez, Conceptualization, Investigation, Methodology, Writing – review and editing | Mónica P. Cala, Formal analysis, Funding acquisition, Investigation, Methodology, Resources, Writing – review and editing | Alejandro Reyes, Conceptualization, Funding acquisition, Investigation, Methodology, Project administration, Resources, Supervision, Validation, Visualization, Writing – review and editing

## DATA AVAILABILITY

All sequences were submitted to the European Nucleotide Archive of the European Bioinformatics Institute under accession number PRJEB67468 and are available at http://www.ebi.ac.uk/ena/data/view/PRJEB67468.

## ADDITIONAL FILES

The following material is available online.

## Supplemental Material

**Supplemental material (mSystems01364-25-s0001.docx).** Supplemental methods, figures, and tables.

## Open Peer Review

**PEER REVIEW HISTORY (review-history.pdf).** An accounting of the reviewer comments and feedback.

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
