## [Reviewer comments · mSystems]

Metagenomic, metabolomic and sensorial characteristics of fermented *Coffea arabica* L. var. Castillo beans inoculated with microbial starter cultures

Maria Madrid-Restrepo, Ana Leon-inga, Aida Peñuela, Monica Cala, and Alejandro Reyes Munoz

Corresponding Author(s): Alejandro Reyes Munoz, Universidad de Los Andes

Review Timeline:

Submission Date:

September 23, 2025

Accepted:

November 19, 2025

Editor: John Gibbons

Reviewer(s): The reviewers have opted to remain anonymous.

Transaction Report:

DOI: <https://doi.org/10.1128/msystems.01364-25>

Re: mSystems01364-25 (Metagenomic, metabolomic and sensorial characteristics of fermented *Coffea arabica* L. var. Castillo beans inoculated with microbial starter cultures)

Dear Dr. Alejandro Reyes Munoz:

Thank you for your patience. One of the previous reviewer's evaluated your revision. I also evaluated the revised manuscript and am satisfied your edits and responses to reviewer concerns. Congratulations, and thanks for submitting to mSystems. I look forward to seeing this in press!

Your manuscript has been accepted, and I am forwarding it to the ASM production staff for publication. Your paper will first be checked to make sure all elements meet the technical requirements. ASM staff will contact you if anything needs to be revised before copyediting and production can begin. Otherwise, you will be notified when your proofs are ready to be viewed.

Sincerely,
John Gibbons
Editor
mSystems

Reviewer #2 (Comments for the Author):

No more comments.